# Role of Type I Interferons during *Mycobacterium tuberculosis* and HIV Infections

**DOI:** 10.3390/biom14070848

**Published:** 2024-07-14

**Authors:** Elsa Anes, José Miguel Azevedo-Pereira, David Pires

**Affiliations:** 1Host-Pathogen Interactions Unit, Research Institute for Medicines (iMed.ULisboa), Faculty of Pharmacy, Universidade de Lisboa, Av. Prof. Gama Pinto, 1649-003 Lisboa, Portugal; miguel.pereira@ff.ulisboa.pt (J.M.A.-P.); dpires@ff.ulisboa.pt (D.P.); 2Center for Interdisciplinary Research in Health, Católica Medical School, Universidade Católica Portuguesa, Estrada Octávio Pato, 2635-631 Rio de Mouro, Portugal

**Keywords:** interferons, tuberculosis, HIV, co-infection

## Abstract

Tuberculosis and AIDS remain two of the most relevant human infectious diseases. The pathogens that cause them, *Mycobacterium tuberculosis* (Mtb) and HIV, individually elicit an immune response that treads the line between beneficial and detrimental to the host. Co-infection further complexifies this response since the different cytokines acting on one infection might facilitate the dissemination of the other. In these responses, the role of type I interferons is often associated with antiviral mechanisms, while for bacteria such as Mtb, their importance and clinical relevance as a suitable target for manipulation are more controversial. In this article, we review the recent knowledge on how these interferons play distinct roles and sometimes have opposite consequences depending on the stage of the pathogenesis. We highlight the dichotomy between the acute and chronic infections displayed by both infections and how type I interferons contribute to an initial control of each infection individually, while their chronic induction, particularly during HIV infection, might facilitate Mtb primo-infection and progression to disease. We expect that further findings and their systematization will allow the definition of windows of opportunity for interferon manipulation according to the stage of infection, contributing to pathogen clearance and control of immunopathology.

## 1. Introduction

Interferons (IFNs) are a broad class of pleiotropic cytokines produced in response to pathogen sensing by innate immune receptors. They were the first cytokines to be discovered, having been described in 1957 as mediators that could “interfere” with “viral replication” in vertebrate cells [1,2,3]. Nevertheless, their effects during non-viral infections, particularly those caused by intracellular bacteria, are now well documented [4,5,6,7,8,9,10,11,12,13].

Two important human pathogens of the 21st century are the human immunodeficiency virus (HIV) and the facultative intracellular bacterial species, *Mycobacterium tuberculosis* (Mtb). HIV emerged as a pandemic in the 1980s, causing an acute infection in humans often characterized by strong flu-like symptoms. This is followed by a chronic infection that results in massive CD4^+^ T-cell depletion by apoptosis and pyroptosis [14], which is known as acquired immune deficiency syndrome (AIDS). The acute infection phase, which lasts between four and eight weeks after transmission, is characterized by a rapid increase in viral load and a subsequent decline in CD4^+^ T cells [15]. Following this initial period, a robust HIV-specific cytotoxic T-lymphocyte (CTL) response develops, effectively suppressing HIV viremia, which marks the transition to the chronic phase [16]. This phase is characterized by an equilibrium between viral replication, viral immune evasion, and the elimination of infected cells by the host immune response [17,18]. More than 1.3 million individuals were acquiring HIV each year, with AIDS resulting in the death of 2 million people annually [19]. Following the introduction of an effective combined antiretroviral therapy, the HIV infection became a long-term chronic condition. Nevertheless, despite the ability to control viral replication, this treatment fails to eradicate the infection [20,21]. Furthermore, in 2022, approximately 9.2 million individuals infected with HIV were not receiving antiretroviral therapy, while 2.1 million individuals undergoing treatment were not virally suppressed [19]. A highly associated opportunistic infection is one with *Mycobacterium tuberculosis* (Mtb), resulting in a syndemic, a situation were co-infection exacerbates the morbidities of each associated pathogen and is considered a major public health concern [22].

*Mycobacterium tuberculosis* (Mtb) is an ancient human pathogen that causes tuberculosis, a respiratory transmissible infection, which evolved by establishing intracellular niches in macrophages [20,23,24,25,26,27]. The infection often progresses through different phases, including primo-infection, latent infection, and finally, the active form of the disease, tuberculosis (TB) [20,22]. The primo-infection may present with relatively mild flu-like symptoms or be completely asymptomatic [28,29]. This stage persists for the first four to eight weeks following transmission and corresponds to the innate phase of the host immune responses. The bacilli replicate in a series of cycles of infection within the macrophages and infection of newly arrived phagocytic cells that internalize live bacteria within apoptotic vesicles [23,24]. The instruction of the adaptive immune responses initiates the progression to a latent, chronic form of the infection (LTB), which is asymptomatic and not transmissible. The bacterial loads remain under control, with replicating bacteria giving rise to dormant bacilli forms [30,31]. Both the primo-innate phase and LTB can be resolved with the complete clearance of the pathogen by the host immune response [28,29]. Among those who are latently infected but do not clear the pathogen, approximately 5 to 10% will evolve to TB within 2 to 10 years [32,33]. In immunocompromised patients, following the primo-infection a direct progression to TB may occur, which can disable the latent phase of infection. 

TB is an active disease characterized by uncontrolled intracellular and extracellular bacterial replication and high necrotic inflammatory events, affecting mostly the lungs and accompanied by severe symptoms that cause high morbidity and mortality worldwide [34,35]. With an estimated global total of 10.6 million people infected in 2022 and 1.13 million TB-related deaths worldwide [36], it is predictable that about 23% of the world population was latently infected with Mtb [37]. Human immunodeficiency virus (HIV) remains a major risk factor for the development of active TB from latency, with 671,000 people estimated to be co-infected with both pathogens, accounting for 167,000 deaths in 2022 [36].

This review focuses mostly on the role of type I IFN during these different stages of infection with Mtb, with HIV, and during the co-infection.

## 2. Origins and Intrinsic Cellular Antimicrobial Roles of Interferons 

To better understand the role of IFN during infections, it may be beneficial to examine the origins of vertebrate evolution, which occurred approximately 450 million years ago [38] (Figure 1). This period is coincident with an expansion of the genome size, extensive virus infections, including those by exogenous and endogenous retrovirus, and the acquisition of retrotransposons and other mobile elements [39]. In fact, jawed fish (e.g., sharks), being predators, were more susceptible to viral infections than jawless early vertebrates. This stimulated the evolution of more efficient and robust immune responses in jawed fish [40]. The endogenization of these mobile elements brought the gene cassettes and provided genome rearrangements that resulted in the biogenesis of innate cytokines, including IFN, as well as the fundamental blocks required for adaptive immunity, namely the class I and class II major histocompatibility complex (MHC-I and MHC-II) molecules, the T- and B-cell receptors (TCR and BCR), and the recombination activating gene (RAG) proteins [40,41] (Figure 1).

IFNs evolved from a class II helical cytokine ancestor, along with the interleukin (IL)-10 cytokine family, and are categorized into three classes: type I (most represented by IFN-α/*β*), type II (IFN-*γ*), and type III (IFN-λs) interferons [42,43,44] (Figure 1). Two rounds of whole-genome duplication (WGD) coordinated with retrotransposition events that occurred between jawless and jawed vertebrates and bony fish [41,45] may have given rise to the two actual loci containing the IL-10 family genes, and potentially the IFN-I and IFN-III loci [46,47]. The origin of the IFN-II gene from the class II helical cytokine ancestor was also an early event in vertebrate evolution [48,49] (Figure 1).

Type I IFNs (IFN-I) are ubiquitously expressed and sensed by all nucleated cells [50], displaying a global range of direct and indirect effects during infection [51]. This includes effects on innate and adaptive immune cells, which are more often associated with inflammatory-related pathogenic consequences [51]. The group is represented by IFN-α and IFN-*β*. The first is produced more strictly but in high amounts by plasmacytoid dendritic cells and by hematopoietic cells (mostly leucocytes), while the second is produced more broadly by fibroblasts, dendritic cells, and epithelial cells [52,53]. A strong antiviral effect is based on the direct effects of IFN-I in activating natural killer cells (NKs) to enhance their cytotoxicity against intracellular pathogens [54,55,56]. An indirect effect is to boost NK immunomodulatory effects evidenced by the secretion of cytokines such as IFN-II, involved in the activation of macrophages to a more microbicidal state [57,58].

Type III interferons or IFN-λs induce strong antiviral effects comparable to those of IFN-I. They are mainly produced by epithelial cells and fibroblasts and are sensed exclusively by mucosal, endothelial, and brain barrier cells. Consequently, in contrast to IFN-I, IFN-III provides local immune protection effects with limited inflammatory responses [59]. 

Type I and type II interferons enhance MHC-I expression in all nucleated cells, enabling better efficacy in presenting viral antigens. In the case of dendritic cells, the induced antigen presentation capacity establishes a bridge for adaptive immunity, thereby cross-priming CD8+ T cells for polarization to adaptive cytotoxic T lymphocytes (CTLs) [58,60,61,62]. CTLs are more pathogen-driven and represent a second strong antiviral effect mediated by interferons to infected cells. IFN-II is represented by IFN-*γ*, which is produced exclusively by immune cells and may be stimulated in response to viral infections. IFN-*γ* possesses limited direct antiviral effects. Conversely, it promotes adaptive and innate responses against intracellular pathogens indirectly through pleiotropic effects on a diverse set of immune cells [51]. IFN-*γ* stimulates the expression of both class I and class II MHC molecules in antigen-presenting cells (APCs) and co-stimulators, thereby promoting the priming of CD4+ T cells into effector T_H_1 cells in addition to CTL polarization [63,64]. IFN-*γ* is secreted from T_H_1, CTLs, and NK cells, which in turn activates macrophages to kill intracellular pathogens in endocytic compartments [63,65]. This process is particularly relevant in eliminating intracellular bacteria. 

Overall, interferons, particularly IFN-I, elicit a cell-intrinsic antiviral state in infected and neighboring uninfected cells through the expression of hundreds of interferon-stimulated genes (ISGs) after signaling from type I interferon receptor (IFNAR) [66] that impairs virus production (Figure 2). A general overview of interferon activation and signaling for ISGs and the roles played by ISGs as positive or negative regulators of inflammation and antimicrobial responses is depicted in Figure 2 (reviewed in [66,67]).

As previously mentioned, they promote the activation of dendritic cells (DCs) [68,69] and NK cells, as well as macrophage function [70], while also enhancing antigen presentation and cellular adaptive immune responses. By facilitating the cooperation of T and B lymphocytes, IFN contributes to the enhanced production of high-affinity and long-lived antibodies [71,72]. IFN-I is more efficacious in the resolution of viral infections, and it also enhances the development of memory immune responses that are capable of responding effectively to future viral challenges [73].

Interferons have emerged in the context of effective clearance of acute viral infections [56], and therefore, their expression and activity are designed to be transient, with a rapid decrease in secretion following the control of viruses. Nevertheless, the protective antiviral effector molecules that are encoded by ISGs can also cause immunopathology during acute viral infections [73,74]. This immunopathology can be observed in several viral infections, including those affecting patients with acute severe lower respiratory infections in hospital intensive care units. These patients display pathological inflammatory responses, yet viral replication is controlled, and the residual IFN-I secretion is already declining [73]. Conversely, they can lead to immunosuppression and loss of virus control during chronic viral infections, in part by increasing the expression of the coinhibitory programmed cell death ligand 1 (PD-L1) and IL-10 signaling [75] (Figure 2).

The role of type I IFNs has been the subject of increasing attention, not only during viral infection but also for their central engagement in the host response to bacterial infections [8,76,77]. The signaling pathway of type I IFN in response to bacterial infections is influenced by many factors. This includes, for example, whether the infecting bacterium is intracellular or extracellular and the subsequent signaling pathways that are initiated. As has been described for viruses, type I IFNs appear to promote or impair bacterial pathogen control and disease pathology depending on the causal species, the virulence of the strain, and the course of an acute or chronic infection [78].

## 3. Interferons during HIV Infection

The mechanisms of the innate immune response have evolved to protect cells from exogenous infectious agents and cellular components released as a consequence of cell injury. Following HIV infection, viral components such as HIV RNA and DNA molecules (ssRNA, dsRNA, RNA:DNA hybrids, and dsDNA) could be detected by PRRs, including toll-like receptors 7 and 8 (TLR7, TLR8), cyclic GMP-AMP synthase (cGAS), and retinoic acid-inducible gene I (RIG-I), which trigger several signaling pathways [79,80] (Figure 2). This allows the activation of distinct cell transcription factors, such as interferon (IFN) regulatory factors (IRFs), NF-kB, and activator protein 1 (AP1), ultimately leading to the upregulation of antiviral and proinflammatory cytokines, including type I interferon IFN-α (referred to here as IFN-I) [81,82]. 

As mentioned before, after being released from the cell, IFN-I binds to IFNAR and activates several intracellular signaling cascades. These culminate in the induction of ISGs [83,84], some of which encode proteins with direct antiviral activity that restrict several steps of HIV replication, such as apolipoprotein B mRNA-editing enzyme catalytic polypeptide 3 (APOBEC3) G/F, tripartite motif 5α (TRIM5α), SAM and HD domain-containing deoxynucleoside triphosphate triphosphohydrolase (SAMHD1), Tetherin, interferon-induced transmembrane protein (IFITM), myxovirus resistance gene 2 (MX2), and Schlafen (SLFN) [85]. Their effects in specific steps of HIV cycle replication are summarized in Figure 3.

IFITMs are primarily located in cellular membranes, including the plasma membrane and endosomal membranes with two transmembrane regions and a cytoplasmic domain, which inversely mirrors the topology of Tetherin [86]. In humans, there are five members of this family of small proteins: IFITM-1, IFITM-2, IFITM-3, IFITM-5, and IFITM-10 [87,88]. 

During HIV infection, IFITM2 and IFITM3 proteins can inhibit the replication of HIV-1 by interfering with the virus’s ability to fuse with host cell membranes, either at the plasma membrane or after endocytosis. Interestingly, although IFITM1 is unable to prevent virus fusion, it suppresses Gag protein expression, contributing to a lower virus production [89]. This suggests that IFITMs may affect more than one step of the HIV replication cycle. In fact, several studies have demonstrated that IFITM proteins are passively incorporated into nascent virions, impairing their capacity to fuse with new target cells and thus reducing their infectivity [90,91,92]. In conclusion, the anti-HIV mechanisms of IFITMs can be described as a two-step model in which IFITMs act not only by protecting target cells from fusion of incoming viruses but also by the production of virions with reduced infectivity. 

TRIM5α inhibits the replication of HIV in the cytoplasm of host cells by binding to specific determinants in the viral capsid (CA) protein. This interaction with the CA protein results in a premature and accelerated uncoating of the HIV capsid, thereby preventing subsequent steps in the viral replication cycle, namely reverse transcription and nuclear import [93]. The dissociation of capsid proteins can occur through either proteasome-independent or proteasome-dependent mechanisms [93,94,95].

The Mx gene encodes two proteins, designated Mx1 and Mx2. During a screening process for antiviral activity of over 380 human interferon-stimulated gene products, an antiviral activity of human Mx2 against HIV was identified [84]. 

Mx2 expression has been demonstrated to potently inhibit HIV-1 infection following reverse transcription but prior integration [96,97,98]. Accordingly, Mx2 may interfere with any one or more of the following replication steps: (i) nuclear import of viral capsid; (ii) uncoating of viral capsid; or (iii) integration of viral double-stranded DNA into the host cell chromosome. An initial observation revealed that Mx2 binds to the HIV capsid in the cytoplasm of the cell, preventing its uncoating through the stabilization of incoming viral capsids [99]. Nevertheless, the interaction with the capsid may be necessary but not sufficient for Mx2 inhibition [100,101,102]. The antiviral action of Mx2 is dependent on the binding of the host cell factor, cyclophilin A (CypA), to specific residues of the CA protein, which constitute the major component of the HIV capsid. CypA is a peptidyl–prolyl isomerase that facilitates the nuclear import of the viral capsid (that encloses the retrotranscribed dsDNA) by guiding it to nucleoporins (Nups) [103,104]. The inhibition of the interaction between CypA and the CA protein abrogates the antiviral activity of Mx2, indicating that it interferes with the function of CypA during nuclear import [98]. The identification of mutations in the CA protein that allow the virus to evade the action of Mx2 [96,97,98,104,105,106,107], in conjunction with the involvement of distinct Nups for nuclear import, depending on the cell type [108], and the precise subcellular localization of Mx2 that is necessary for anti-HIV activity [106,108,109], add further complexity to the antiviral mechanism employed by Mx2 to inhibit HIV replication. 

SAMHD1 was identified as an HIV-1 restriction factor that blocks early-stage virus replication in dendritic cells (DCs) [110,111]. It acts by depleting the intracellular pool of deoxynucleoside triphosphates (dNTPs), thus impairing HIV reverse transcription and productive infection [112,113,114]. This mechanism results in a dual effect of SAMHD1 on DCs. On the one hand, it renders them less permissive to HIV infection. On the other hand, it enables HIV to avoid the cytosolic sensing of viral nucleic acids that would otherwise trigger IFN-mediated antiviral immunity [115,116].

SAMHD1 is antagonized by Vpx, an accessory protein encoded by the *vpx* gene, which is present in SIVsm/SIVmac and HIV-2. This is achieved by targeting it for proteasomal degradation. Consequently, the degradation of SAMHD1 in HIV-2-infected dendritic cells (DCs) renders them much more susceptible to viral replication, allowing for the faster accumulation of full-length viral DNA [117].

The APOBEC3G protein is incorporated into the HIV virion following a specific interaction with the amino-terminal region of the nucleocapsid (NC) domain of the HIV Gag polyprotein during the assembly of the new viral particle [118,119,120,121,122]. 

In the early stages of reverse transcription, shortly after the viral genome is introduced into a new host cell, the APOBEC3G enzyme begins to modify the minus strand of viral DNA. The process, called cytidine deamination, involves the conversion of deoxycytidines (dCs) into deoxyuridines (dUs). This results in a change at the DNA level, potentially altering the sequence of the viral genome [123,124]. This results in a dG to dA hypermutation in the HIV-1 double-stranded DNA genome of the replicating virus, which ultimately leads to mutations and stop codons that disrupt the normal expression and function of viral proteins [125,126].

This potent inhibitory effect on HIV replication is observed exclusively in HIV mutants lacking a functional vif gene. This is because the Vif protein blocks the function of APOBEC3G [123,127,128,129,130,131,132]. The Vif protein circumvents this mechanism of HIV replication inhibition by binding to APOBEC3G in the cytoplasm of infected cells prior to virion assembly. This binding directs APOBEC3G for polyubiquitination and proteasomal degradation, preventing its inclusion into the newly formed virions [123,129,130,133,134].

Schlafen (SLFN) proteins are involved in various cellular processes, including cell proliferation, differentiation, and immune responses [135,136]. In the context of HIV infection, SLFN proteins play significant roles in the innate immune response against the virus, exhibiting direct antiviral activity. For example, SLFN11 has been shown to restrict HIV replication by targeting the synthesis of viral proteins [137]. It achieves this by interfering with the translation of HIV mRNA, thereby inhibiting the production of HIV proteins necessary for viral replication. This mechanism relies on the binding of SLFN11 to cellular tRNAs, reducing its pool [137], thus counteracting the observed HIV-driven enrichment of specific tRNAs decoding A-ending codons, required for the expression of the RNA genome of HIV that shows an extremely high frequency of adenosine at the third position of the codon [138].

Finally, Tetherin is a type II membrane protein that is primarily located in cholesterol-rich microdomains, also known as lipid rafts. It is highly expressed at the plasma membrane of B cells, bone marrow CD34+ cells, and T cells [139]. This restriction factor acts by inhibiting the egress of HIV and other enveloped viruses by tethering mature virions to the host cell membrane [140,141,142,143,144,145,146,147].

As a well-adapted human pathogen, HIV has evolved a number of mechanisms to avoid the detection of its genome by the host’s PRR. The most relevant of these mechanisms is the conversion of the RNA genome into DNA within the viral capsid. In fact, the process of descapsidation seems to occur at the nuclear pore or even inside the nucleus of the infected cell [148,149,150,151,152]. The maintenance of the capsid structure is important for nuclear import and integration of the viral genome, as well as for creating a protective shield, which reduces the amounts of HIV RNA, HIV DNA, and intermediate molecules that could be detected by cytoplasmic PRRs. Furthermore, HIV has evolved additional mechanisms that counteract the restriction factors induced upon IFN production, primarily through viral proteins such as Vif, Nef, Vpu, and Vpx [153,154] (Figure 3). It is noteworthy that different subtypes of IFN-I demonstrate varying degrees of anti-HIV activity, particularly in the later stages of viral replication, indicating a hierarchy of antiviral potency [155]. 

During the acute phase of infection, which lasts for a few weeks after transmission, HIV triggers a cytokine storm characterized by the upregulation of several cytokines and chemokines, including IFN-I, IL-15, IL-6, TNF-α, IL-8, IL-18, and IFN-*γ* [82]. A strong IFN-I production is crucial in controlling initial HIV infection and establishing the viral set point. Interestingly, the HIV variants responsible for new infections after transmission (known as transmitted/founder viruses) exhibited reduced susceptibility to IFN-I antiviral activity, which provides a selective advantage in the context of the transmitted viral population as well as after antiretroviral interruption [156,157]. 

In the chronic phase of HIV infection, after seroconversion, a clear and constant production of inflammatory mediators, including IFN-I, is detected in the plasma and lymph nodes [158]. This leads to chronic immune activation and systemic inflammation observed in HIV-infected individuals, even in those under antiretroviral therapy [159]. This chronic inflammation is, at least in part, responsible for the depletion and dysfunction of T-CD4+ lymphocytes during the chronic HIV infection. In fact, an analysis of HIV-infected patients shows that higher viral loads and increased rates of disease progression correlate with overexpression of ISGs [160,161,162]. It is noteworthy that, in several non-human primates infected with simian immunodeficiency virus (SIV), although all mount a robust IFN-I response during acute infection, the key factor distinguishing pathogenic from non-pathogenic SIV infection is the resolution of this IFN-I response in a short period of time [163]. In addition, IFN-I signaling promotes the activation of T-CD4+ lymphocytes and the upregulation of CCR5 (a major coreceptor for HIV entry into target cells), which results in increased availability of target cells and a subsequent enhancement of HIV replication. This perpetuates a cycle of immune activation and IFN signaling that contributes to cell death through several mechanisms, including apoptosis mediated by TNF-α-related apoptosis-inducing ligand (TRAIL) [164].

In addition to T-CD4+ lymphocyte depletion, IFN-I also affects cell-mediated immune responses by indirect mechanisms. As previously stated, these include the production of immunosuppressive cytokines such as IL-10, or the increased expression of ligands PD-1 and PD-L1 [165]. In conjunction with the continuous activation of T cells observed during HIV infection, both mechanisms result in the exhaustion of these cells. This, in turn, compromises the ability to control other pathogens, including those that are able to establish a chronic, long-lasting infection, such as Mtb. 

## 4. Interferons during *Mycobacterium tuberculosis* Primo-Infection and LTB Infection

As previously stated, infection with Mtb may progress through distinct stages. The initial primo-infection is followed by the latent infection (LTB), going from a mild to asymptomatic phase during which bacillary replication is controlled. This may or may not result in the total clearance of the pathogen in a healthy host. If the host fails to eliminate the pathogen, LTB can progress to tuberculosis, the active disease. This is a phase of the infection that results in a strong inflammatory pathology and uncontrolled bacterial replication. Distinct immunocompetent responses of the host are observed during these phases, and likewise, the role of type I IFNs may vary from protective, allowing the control of the infection, to detrimental, contributing to disease progression. Their detrimental role during the progression to tuberculosis is well documented, with the interferon signature, as indicated by the expression of ISGs, being highly detected in patients with tuberculosis, in contrast to those with LTB [166,167,168]. 

During the phases of Mtb infection when the host is still immunocompetent and able to control the bacterial loads, the data available regarding the role of interferons are quite limited, and mostly based on results from in vitro or ex vivo models of TB. The pathogen is transmitted through inhaled droplets containing a few bacilli in the respiratory tract. The first cells to encounter the bacilli following aerial transmission are the alveolar macrophages. These are permissive cells and allow the establishment of intracellular niches within phagosomes. Basal levels of IFN-I expression by these cells result in the recruitment of more permissive myeloid cells, which will internalize free bacteria as well as live intracellular bacilli from dying apoptotic cells. This allows for the controlled expansion of the bacilli within an innate granuloma [22,23]. Dendritic cells, which are professional APCs, migrate to draining lymph nodes, where they present Mtb antigens to initiate the adaptive immune response. Effector T lymphocytes will migrate to the site of infection, licensing macrophages to enhance their ability to kill the pathogen. Altogether, these processes allow the formation of an adaptive granuloma, which contains the bacillary load in the lungs [22]. A continuous supply of effector T-CD4+ lymphocytes to the granuloma is essential for maintaining the structure and control of the infection. 

The protective effect of cytokines in the axis IL-12/IFN-*γ*/TNF-α and IL-1*β* to control the infection is evident (Figure 4). IL-12 is produced in response to innate immune pattern recognition receptors (PRRs) such as the transmembrane toll-like receptors (TLRs), which sense Mtb-associated molecular patterns (PAMPs) by APCs. This promotes the differentiation of T_H_1 cells and IFN-*γ* production. IFN-*γ* further activates Mtb-infected macrophages to secrete TNF-α and other protective cytokines, thereby enhancing their microbicidal responses through the production of reactive oxygen and nitrogen species [63,65]. 

A basal level of autocrine type I IFN signaling has been correlated with high-level production of protective IL-12 and TNF-α by macrophages in response to Mtb infection [169] (Figure 4). During the initial phases of infection, the microbiota in the aerial mucosae plays a crucial role in inducing basal levels of IFN-I [170], which leads to the development of an attendant tonic IFNAR signaling and equips immune cells to rapidly mobilize effective antimicrobial programs [66]. Consequently, the host’s immune activation status for the basal level of IFN-I at the moment of transmission may prevent the establishment of the infection.

Similarly, the innate immune sensing of Mtb PAMPs induces the secretion of IL-1*β* through the activation of inflammasome complexes that further enhance these protective responses [23]. ESAT-6 has been described as a major PAMP inducing the NLRP3 inflammasome and IL-1*β* secretion [171]. However, an exaggerated secretion of this cytokine will display strong pathological inflammatory effects through high proinflammatory cytokine exacerbation and high local recruitment of neutrophils [172,173,174]. It was hypothesized that Mtb evolved to counteract these pathological effects by inducing type I [77,175,176] and type II IFNs [173]. It is, therefore, not surprising that IFN-I is indeed induced in response to ESAT-6, which is part of the ESX-1 protein secretion system [177,178]. This type VII secretion system is responsible for the delivery of several Mtb PAMPS from phagocytic vesicles into the cytosol, including bacterial DNA [23,171]. The cytosolic PRRs, including the cGAS-cGAMP-STING pathway, highly expressed in myeloid cells, are capable of sensing microbial DNA, leading to the production of IFN-I [23,178] (Figure 2).

IFN-*γ* primarily targets macrophages, resulting in the production of nitric oxide (NO) through the activation of the inducible nitric synthase iNOS [63,179]. The direct effect of NO is to decrease the NLRP3 inflammasome responses and to limit the secretion of IL-1*β* [173]. Consequently, restricted recruitment of neutrophils and TNF-α secretion will contribute to the control of the infection with homeostatic effects for the host [173,174]. Another effect of IL-1*β* is the induction of cyclooxygenase-2 (COX) activity, which results in the secretion of prostaglandin E2 (PGE2). This eicosanoid protects the inner mitochondrial membrane, preventing infected cells from undergoing necrotic cell death while promoting apoptosis [180]. During the asymptomatic phases of infection, this process contributes to the control of the bacillary load in infected macrophages within granuloma structures, while also contributing to prevent the exacerbation of inflammation [181]. 

Ex vivo models of TB demonstrate that type I IFN exerts a more pronounced inhibitory effect on IL-1*β* than IFN-*γ*. IFN-I can act directly by inhibiting the transcription of pro-IL-1*β* [182] or indirectly by inducing IL-10, which downregulates the expression of pro-IL-1*β* [183,184]. Another inhibitory effect occurs by targeting COX, which counteracts the effects of IL-1*β* [180]. Other signaling pathways initiated by IFN-I result in an inhibitory effect on the NLRP3 inflammasome, thereby reducing IL-1*β* maturation [185]. 

A balance between type II and type I interferons allows the fine-tuning of IL-1*β* production to successfully control the bacillary loads while restricting inflammatory responses, crucial for maintaining the asymptomatic phases of Mtb infection.

Finally, IFN-I contributes to the control of the bacillary load and the inflammatory responses by reprograming macrophage metabolism [186]. Basal levels of type I interferons released by cells at the infected tissue interface, including infected macrophages, are sensed by IFNAR in an autocrine or paracrine manner, inducing the immunoresponsive genes (IRGs) through the positive transcription factor IRF1 [187]. IRG1 is a mitochondrial enzyme that produces the metabolite itaconate by decarboxylating cis-aconitate, a tricarboxylic acid (TCA) cycle intermediate [188] (Figure 4). 

Both IRG1 and itaconate have demonstrated several antimicrobial [188,189] and immunoregulatory functions [190,191,192,193]. IRG1 induces the use of fatty acids for oxidative phosphorylation (OXPHOS) and the production of mitochondrial reactive oxygen (mtROS), thereby improving macrophage bactericidal activity towards intracellular bacteria [189]. The antimicrobial activity of itaconate is attributed to its inhibition of two key enzymes in the metabolism of Mtb, namely isocitrate lyase in the glyoxylate shunt and methylcitrate lyase in the methylcitrate cycle (Figure 4). These enzymes are required for the survival of pathogens in glucose-limiting conditions found in phagosomes [188,194,195,196,197]. During innate cytokine signaling, the juxtaposition of mycobacteria-containing phagosomes with mitochondria allows for the transfer of itaconate between compartments [198]. 

In addition to its potential antibacterial properties, both IRG1 and itaconate are involved in the control of inflammation caused by excessive neutrophil infiltration during Mtb infection [199]. Indeed, continuous production of itaconate has been demonstrated to suppress the production of inflammatory cytokines, including IL-1*β*, IL-6, and IL-12, as well as mitochondrial ROS, by infected macrophages [192,193]. This capacity to modulate inflammation could be of relevance, as an excessive immune response can prevent Mtb clearance and cause pathology [174,200].

## 5. Interferons during *tuberculosis*

Several cohort and meta-analysis studies have identified a type I IFN signaling signature in tuberculosis [166,168,201,202,203,204]. These findings indicate that high IFN-I signaling play a detrimental role in the progression from chronic LTB to active disease (Figure 4). As previously stated, IFN-I is strongly induced to clear pathogens during the acute infection phase, which is followed by a fast decline in its secretion and the consequent return to a healthy homeostatic state. Alternatively, the infection may progress, resulting in the death of the patient. But what occurs when the infection becomes chronic, and IFN-I is continuously produced? During the acute phase of influenza virus infection, a short-duration disease, the plasma concentration of IFN-I increases until it reaches approximately 10^5^ fg/mL, with activities averaging approximately 10^2^ IU/mL [205]. Conversely, during LTB or TB infection, the plasma concentrations and activities of IFN-I are below 5 fg/mL and IU/mL, respectively [206]. However, an examination of the transcriptomic profile of ISGs reveals a signature of IFN-I in peripheral blood leukocytes; consequences become evident, highly correlated with the extent of lung pathology [204,207]. This allows the distinction between active disease and LTB [206]. These include the overexpression of IFITs, MX1, OAS1, IRF1, and ISG15 [204,207,208]. The downstream immune cellular responses to IFN-I signaling, including the overexpression of IL-10 and PD-L1, are at least in part mediated by type I IFN-activated STAT3 that leads to the repression of signals for proinflammatory cytokines [66] (Figure 2). The overexpression of ISG15 is observed in human monocytes during tuberculosis, and it plays a role as a cytokine that induces IL-10 secretion [208]. Moreover, neutrophils and monocytes isolated from patients with TB revealed a high ISG signature, in contrast to T lymphocytes, indicating a detrimental role of these cells in disease progression [166,209]. In addition, inadequate control of Mtb infection in various TB models is associated with an extensive neutrophil response in the lungs [172,210]. The low concentration of IFN-I observed in plasma from TB patients leads to the hypothesis that a high local secretion occurs at the site of infection. In fact, interstitial macrophages and plasmacytoid dendritic cells were found in granulomas as major producers of IFN-I during active TB [6]. Surprisingly, it was found that the predominant activator sources were the DNA-containing neutrophil extracellular traps (NETs) [6].

The increased local secretion of IFN-I during TB results in the accumulation of immunosuppressive effects, which are mediated by the high expression of PD-1 and PD-L1 coinhibitory ligands and concomitant increase in IL-10 secretion [169,180]. Unbalanced IL-10 can counteract the protective effects of IFN-*γ* (Figure 4) and other protective cytokines, as well as interfere with IL-1*β* production via inflammasome, which can result in uncontrolled bacterial replication [180] (Figure 5). An emerging hypothesis suggests that IFN-I-related loss of effective control of Mtb may be attributed to its interference with IL-1*β*-controlled eicosanoid lipid mediators, such as PGE2 [181]. The ratio of host-protective PGE2 over host-detrimental 5-lipoxygenase products such as lipoxin A4 (LXA4) and leukotrienes (LTB4) is enhanced by IL-1*β* signaling [180] (Figure 5). Type I IFN signaling counterbalances this process, resulting in increased levels of both LXA4 and LTB4 while concomitantly limiting PGE2 production [167]. In contrast to PGE2, both LTB4 and LXA4 induce mitochondrial stress, leading to organelle damage and necrotic cell death. Given that mitochondria have an endosymbiotic origin and retain vestiges of their bacterial past, damaged mitochondria release potent PAMPs, including their DNA (mtDNA), which activates the nucleic acid sensor cGAS. This signals through the adaptor protein STING to induce robust IFN-I production, thereby exacerbating their detrimental effects [178]. Furthermore, LTB4 and LXA4 are potent chemoattractants for neutrophils and LTB4, with an exacerbated effect on TNF-α secretion [166]. The direct consequence of this is an increase in necrotic cell death of Mtb-infected macrophages or neutrophils. This, in turn, exacerbates uncontrolled bacterial replication and pathological inflammation that characterizes the disease [200,211,212,213] (Figure 5).

## 6. Interferons during HIV-Mtb Co-Infection

In the context of HIV-Mtb co-infection, it is essential to considerer IFN-I as a dichotomous cytokine [70,214,215]. As stated, during the initial stages of Mtb infection, IFN-I exerts beneficial effects through the activation of immune cells and protective immune responses. However, the sustained production of IFN-I, as observed during chronic HIV infection in humans or pathogenic SIV infection in non-human primates, leads to immunosuppressive responses including high levels of IL-10 secretion and T-cell exhaustion and dysfunction, hindering effective control of the Mtb infection and the exacerbation of detrimental effects to the host. 

The effects of a pre-existing HIV infection on the infection, survival, and proliferation of Mtb within the lung parenchyma must be regarded as a complex network of factors. Although difficult to ascertain, the establishment of a new Mtb infection has been shown to occur more frequently in HIV-infected patients than in non-infected individuals [216]. Nevertheless, it has been demonstrated that in the context of co-infection there is an impaired ability of Mtb-infected alveolar macrophages to acidify phagosomes containing Mtb bacilli, together with deficient Mtb growth control by neutrophils within the lungs [217,218]. This latter finding was attributed to an impairment of CD16- and CD35-mediated opsonophagocytosis of Mtb by neutrophils [219]. It is therefore plausible that the presence of a pre-existing HIV infection may influence the ability of the innate immune response to control Mtb during the initial stages of infection following transmission, thereby increasing the probability of establishing an Mtb infection. Furthermore, the effects of high IFN-I long-lasting signaling induced by an established HIV infection may contribute to the rapid development of an Mtb primo-infection directly to active TB, overcoming LTB. 

Although IFN-I can enhance the ability of neutrophils to clear Mtb infection by several mechanisms, including the enhancement of their phagocytic and increased microbicidal activity [215,220], an existing HIV infection can affect neutrophil function in different ways [221]. A number of studies have demonstrated that neutrophil activation, phagocytic capacity, and microbial killing mechanisms are negatively impacted in HIV-infected individuals [217,222,223,224]. These functional abnormalities are directly correlated with high levels of viremia [217,225] and lower CD4+ T-cell counts [224]. They also increased with disease progression and are improved with antiretroviral therapy [217,224,225,226]. Furthermore, HIV infection can result in reduced neutrophil counts (neutropenia) due to a combination of decreased production in the bone marrow, increased destruction in the bloodstream, and sequestration in lymphoid tissues [227,228]. Alternatively, HIV infection can also induce an altered expression of CD16 and CD35 either directly or indirectly through dysregulated cytokine production [219,229,230,231,232,233,234]. In conclusion, HIV infection can impair neutrophil function through a number of mechanisms, including direct viral effects, immune dysregulation, and decreased production. These impairments contribute to the immunocompromised state that is characteristic of HIV-infected individuals, thereby increasing their susceptibility to subsequent infection by Mtb.

During latent Mtb infection, the most evident factor contributing to the diminished control of Mtb replication is the depletion of T-CD4+ lymphocytes induced by HIV [235], including T_H_1, and consequently a depletion of CTLs that are dependent on the cytokines of the last to become effector cytotoxic cells. Both are crucial for controlling Mtb infection. However, additional factors must be present, as patients undergoing antiretroviral therapy (ART) with high levels of T-CD4+ lymphocytes in the peripheral blood continue to exhibit an elevated risk of active tuberculosis [216,236]. Furthermore, IFN-I was found to impair the influx of T_H_1 cells to the lungs, which is associated with the development of immunopathology during viral co-infections [237]. 

A defining feature of Mtb infection is the formation of a tuberculous granuloma within the lungs soon after transmission. Granulomas are organized structures formed during the immune response to Mtb infection, and they provide a mechanism for containing Mtb [238,239]. The smoldering levels of IFN-I production during HIV infection have both beneficial and detrimental effects on the control of Mtb infection within the granuloma. These effects appear to be strain-dependent, influenced by various factors such as the stage of HIV infection, host immune status, and the cytokine microenvironment within the granuloma [56,214]. More precisely, experimental mouse infection with hypervirulent Mtb strains demonstrated an increased virulence in correlation with increased levels of IFN-I [176,240,241]. However, this correlation was not universally observed [242]. The levels of proinflammatory cytokines, namely IL-1α and IL-1*β*, are also crucial for the control of Mtb infection, and their production is influenced by IFN-I [176,240,241,243]. Finally, the maintenance of the granuloma structure is also affected by peripheral blood counts of T-CD4+ lymphocytes. In the late stages of HIV infection and advanced immune suppression, the granuloma structure is disrupted, accompanied by extensive necrosis, a significant increase in neutrophil infiltration, and the release of Mtb bacilli [244,245,246]. 

It can be concluded that the reported beneficial effects of IFN-I are scarce and contradictory [247,248,249], and point mainly to a protective role against the development of active tuberculosis in the absence of IFN-*γ* [250,251]. Conversely, as previously stated, IFN-I signaling appears to contribute to the dysregulation of the inflammatory response, which can lead to immune pathology, granuloma disintegration, and the exacerbation of disease [6,168,176,177,210,214,252,253]. 

In the context of HIV-Mtb co-infection, it is well established that Mtb facilitates HIV replication from viral sanctuary niches, thereby contributing to the progression of HIV to AIDS [254]. The contribution of latent tuberculosis to HIV progression is indeed supported by evidence indicating that Mtb-induced immune activation and changes in the inflammatory milieu at infected sites have systemic effects [255]. In fact, it has been demonstrated that during Mtb infection, immune responses contribute to the increased replication of HIV-1 in the blood [256] and at sites of bacterial infection in the lungs [257]. Mtb virulence factors, such as the wall glycolipid LAM, have been demonstrated to induce the secretion of proinflammatory cytokines, including TNF-α [258]. This, in turn, activates transcription factors such as AP-1 and NF-κB in CD4+ T lymphocytes and macrophages that contain proviral DNA. This results in the transcriptional activation of HIV long terminal repeats (LTRs) and the production of new viral particles [259,260,261].

## 7. Conclusions

Type I interferons emerged in the context of acute infections with potent antiviral effects sometimes associated with the induction of detrimental inflammatory consequences. HIV and Mtb infections have in common a primo-infection that progresses to a chronic phase. During the acute phase of HIV infection, a cytokine storm is observed, accompanied by elevated levels of IFN-I secretion. In the case of Mtb, this innate, asymptomatic acute phase is often accompanied by tonic levels of IFN-I that are protective and control bacterial loads. During the chronic phases, basal levels of IFN-I may persist for an extended period of time. However, the progressive increase in expression of IL-10 and of the coinhibitors PD-1 and PD-L1 leads to immunosuppression and T-cell exhaustion. The presence of a high level of the IFN-I signature is a biomarker for the progression from LTB to TB. During the chronic phase of HIV infection, after seroconversion, a clear and sustained production of inflammatory mediators, including IFN-I, is detected. This results in chronic immune activation and systemic inflammation, which are, at least in part, responsible for the depletion and dysfunction of T-CD4+ lymphocytes that lead to AIDS. Regarding Mtb infection of individuals infected with HIV, the presence of this immunosuppressive environment may facilitate the fast progression to tuberculosis.

The potential of IFN as a therapeutic agent for HIV or Mtb infection is a topic of ongoing debate, with opinions divided on whether its effects are beneficial or detrimental. Recent experimental data permit the consideration of IFN therapies in specific contexts. For example, the administration of IFN-α during BCG vaccination has been shown to promote the production of protective cytokines (IFN-*γ*, TNF-α, IL-12) and to provide increased protection against Mtb infection compared to BCG alone [262]. Infection with an Mtb strain resistant to rifampicin and in the absence of IL-1 signaling demonstrated a protective effect of IFN-I. This was likely due to the induction of nitric oxide, which restricted Mtb replication, lung pathology, and neutrophil infiltration [263]. Indeed, the opposing effects driven by IFN-*γ* have led to the conclusion that this is a promising therapeutic option for the treatment of multidrug-resistant tuberculosis, which is worthy of further studies [264]. 

Further research is required in this field to better clarify the mechanisms by which the immune response can be harnessed to the greatest benefit of the host, while minimizing any adverse effects. Additionally, the development of therapeutics that target interferon signaling for the purpose of infection control represents a promising avenue for future research.

## Figures and Tables

**Figure 1 biomolecules-14-00848-f001:**
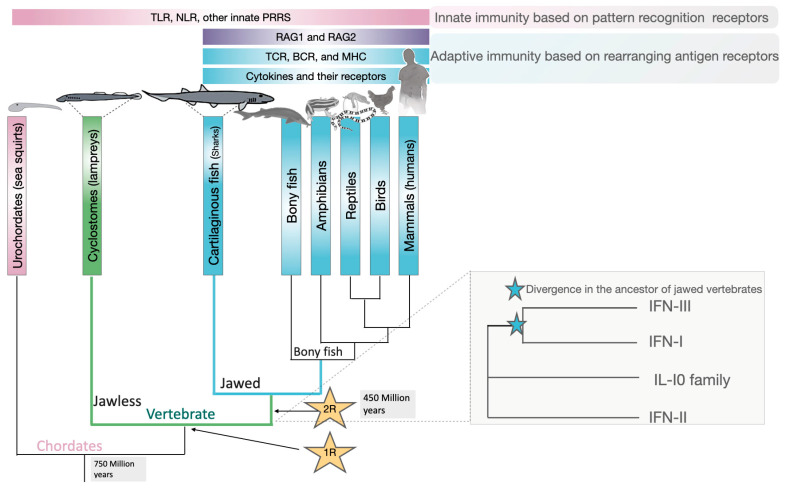
The evolution of vertebrates from jawless to jawed species occurred concurrently with two rounds of whole-genome duplications. The first (star labeled 1R) and the second (star labeled 2R) are estimated to have occurred approximately 450 million years ago. This genome expansion resulted in the incorporations of cassettes of adaptive immunity, along with cytokines and their receptors. One hypothesis suggests that IFNs originated within the class II α-helical cytokine family, where antiviral interferons form a clade along with the interleukin (IL)-10 cytokine family. All three IFN classes were present in the jawed vertebrate ancestor.

**Figure 2 biomolecules-14-00848-f002:**
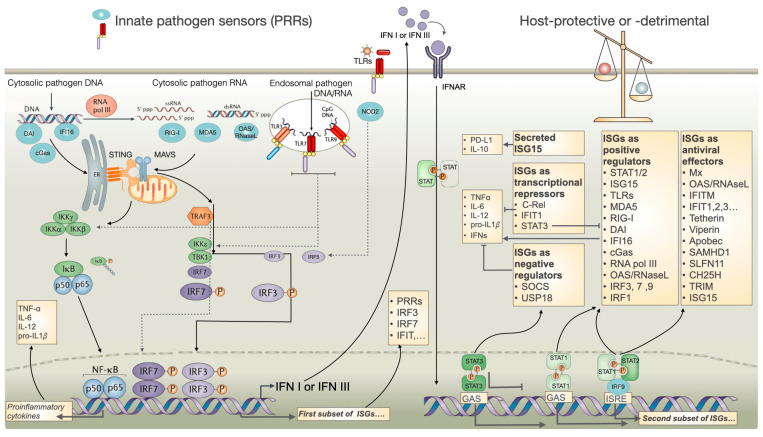
Schematic representation of innate pattern recognition receptors (PRRS) and the subsequent activation of IFN and ISGs. Cytoplasmic or endosomal membrane-associated toll-like receptors (TLRs) or cytosolic-associated immune sensors that recognize pathogen-associated molecular patterns (PAMPs) initiate a signaling cascade that results in the secretion of IFN, ISGs, and proinflammatory cytokines. Gamma interferon-inducible protein 16 (IFI16), DNA-dependent activator of IFN regulatory factors (DAI), and cyclic GMP-AMP synthase (cGas) are sensors for DNA. Retinoic acid-inducible gene 1 (RIG-I), melanoma differentiation-associated protein 5 (MDA5), and oligoadenylate synthetase (OAS) and latent endoribonuclease (RNaseL) recognize foreign RNA. The signaling proceeds to transcription factor activity by stimulator of IFN genes (STING) and mitochondrial antiviral-signaling protein (MAVS) at the ER/mitochondrion interface. Consequently, this results in the activation of interferon (IFN) response factors 3 or 7 (IRF3/7), or alternatively, the activation of NF-κB. These transcription factors translocate to the nucleus, where they activate specific promoters, triggering the expression of IFN and a first subset of ISGs. IFN is secreted out of the cell and signals through IFNAR, which is then transmitted to signal transducers and activators of transcription (STATS). This results in the expression of a large spectrum of ISGs with distinct functions, including antiviral/antimicrobial effectors, and negative (repressors of inflammation) or positive (inducers of inflammatory responses) regulators of IFN signaling. The disabling of the interferon responses may be induced by ISGs, such as the suppressor of cytokine signaling (SOCS) or ubiquitin carboxy-terminal hydrolase 18 (USP18), which block the signals directly from IFNAR or TLRs. In addition, they function as transcriptional repressors of proinflammatory cytokine transcription factors, such as C-Rel and IFIT1. STAT3 may act as a transcriptional repressor by sequestering STAT1. Secreted ISG15 stimulates an increase in the expression of immunosuppressive cytokine IL-10 and the programmed cell death ligand 1 (PD-L1) in monocytes. The equilibrium of these pathways may result in either protective or detrimental effects on the host. Gamma-activated sequences (GASs); interferon-stimulated response element (ISRE); IFN-inducible transmembrane (IFITM); IFN-induced protein with tetratricopeptide repeats (IFIT); virus inhibitory protein, endoplasmic reticulum-associated, IFN-inducible (Viperin); myxovirus resistance (Mx); cholesterol-25-hydroxylase (CH25H); tripartite motif protein 5α (TRIM5α); apolipoprotein B mRNA-editing enzyme catalytic polypeptide 3 (APOBEC3); HD domain-containing deoxynucleoside triphosphate triphosphohydrolase (SAMHD1).

**Figure 3 biomolecules-14-00848-f003:**
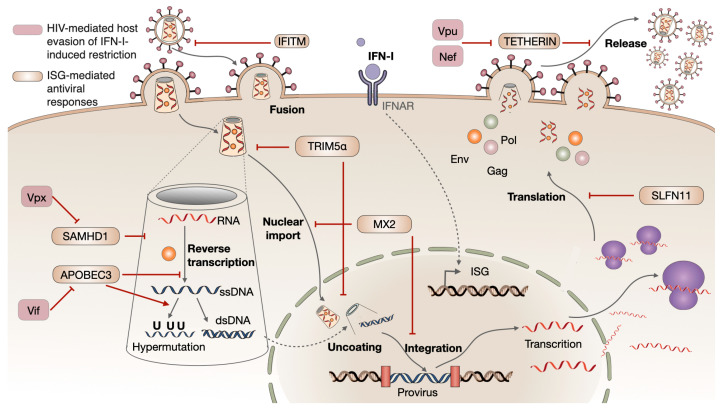
Interferon-mediated antiviral mechanisms against HIV infection affect all stages of the viral cycle, from fusion, uncoating, reverse transcription, nuclear import, and translation, to virus particle release. HIV developed evasion mechanisms towards these IFN-I-induced restriction factors. APOBEC: apolipoprotein B mRNA-editing enzyme, catalytic polypeptide-like; IFITM: interferon-induced transmembrane protein; MX2: myxovirus resistance protein 2 (also known as MX dynamin like GTPase 2); SAMHD1: sterile alpha motif [SAM] and histidine/aspartic acid [HD] domain-containing protein 1; TRIM5α: tripartite motif 5 alpha.

**Figure 4 biomolecules-14-00848-f004:**
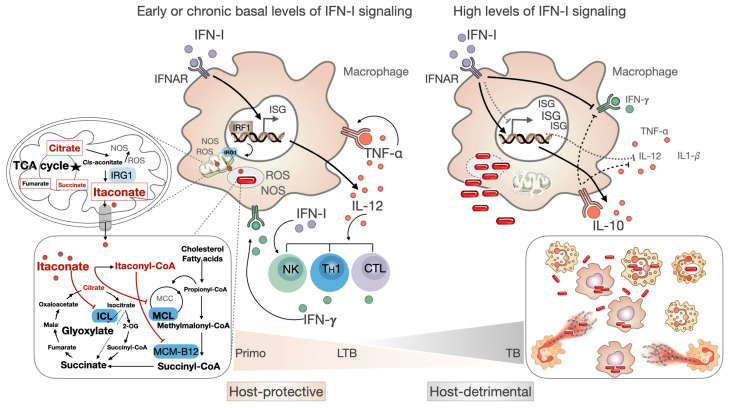
Role of IFN-I during infection with Mtb. Basal levels of interferon signaling are protective for the host. Signaling via the IFNAR receptor results in the expression of hundreds of ISGs. Among these are the protective cytokines IL-12 and TNFα. The latter protects the host in an autocrine or paracrine manner, leading to the activation of the infected macrophage to a more bactericidal state. IFN-*γ* produced from IL-12-stimulated lymphocytes further increases the oxidative burst. Itaconate, produced by the immunoresponsive gene 1 (IRG1) from cis-aconitate, modulates the TCA cycle by regulating succinate dehydrogenase activity for fumarate production. During Mtb infection, first, itaconate induces mtROS and inducible nitric oxide synthase (iNOS), leading to pathogen control. Itaconate has additional antimicrobial activity that inhibits methyl citrate lyase (MCL) in the methyl citrate cycle (MCC) and isocitrate lyase (ICL) in the glyoxylate shunt, enzymes that are essential for Mtb survival. Moreover, itaconyl-coenzyme A (CoA) targets B12-dependent methylmalonyl-CoA mutase (MCM-B12), thereby inhibiting bacterial growth. High IFN-I signaling is observed during the progression of tuberculosis, with detrimental effects on the host. These effects are mainly due to the increased secretion of the immunosuppressor IL-10, a decreased secretion of protective cytokines, and opposing the protective effects of IFN-*γ*. This environment of chemoattractant signals attracts myeloid cells to the granuloma, which allows an uncontrolled bacterial replication concomitant with high levels of inflammatory cell death.

**Figure 5 biomolecules-14-00848-f005:**
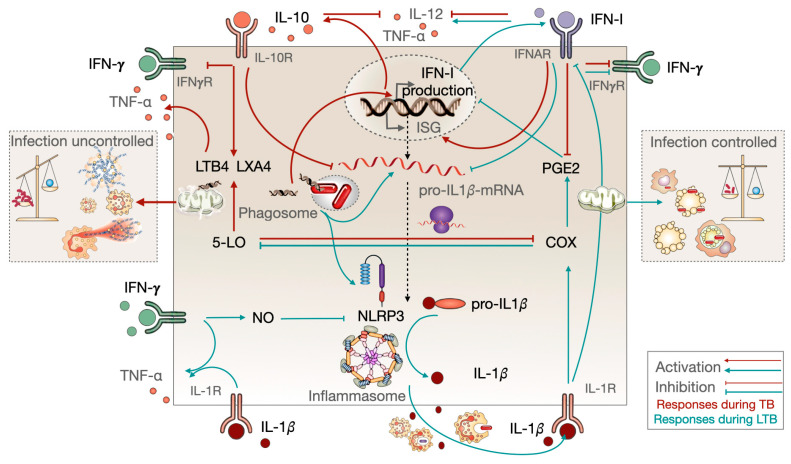
Immune responses during LTB and during progression to TB. Red lines represent the responses during active disease with high IFN-I signature signaling. This leads to the inhibition of IL-1*β* secretion, uncontrolled bacilli replication, and strong inflammatory responses through massive neutrophil recruitment and NETosis, which is followed by the stimulation of 5-LO products. These mechanisms culminate in lung cavitation and the eventual death of the patient in the absence of antibiotic treatment. The green lines represent the events that occur during LTB, which promote infection control through a series of phagocytic events and apoptotic cell death mediated by the stimulation of COX products; in contrast, IL-1*β* secretion is kept under protective, balanced levels.

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
