# Peer review of "Role of Type I Interferons during Mycobacterium tuberculosis and HIV Infections"

_biomolecules, 2024, doi:10.3390/biom14070848_

Round 1
Reviewer 1 Report
Comments and Suggestions for Authors
The manuscript biomolecules-3088345 titled “Role of type I interferons during Mycobacterium tuberculosis and HIV infections” by Elsa Anes et al., is a review focused on summarizing the distinct roles of type I interferons in the acute and chronic stages of both tuberculosis and HIV infection, and importantly, the mechanisms by which type I interferons can contribute to Mtb infection in HIV-infected individuals and to disease progression during Mtb-HIV coinfection. The review draws insight from 176 references, including the latest works on the topic, it is clearly written, easy to read and to understand, and it represents a valuable contribution to the field. The previous reviews I have found, discussing the role of type I interferons in Mtb-HIV coinfection (du Bruyn et al., Current Opinion in HIV and AIDS 13(6):p 455-461, November 2018., DOI: 10.1097/COH.0000000000000501 and Ahmed et al., Oral Dis. 2016 Apr;22 Suppl 1:53-60. doi: 10.1111/odi.12390. PMID: 27109273.) highlight several mechanisms discussed in the current review but neither discusses the topic in such detail or contain works as recent as the manuscript of Anes et al. In conclusion, I recommend the acceptance of the current manuscript after minor revision.
My notes to the authors:
1) I feel a slight break in the flow of the paper between the tuberculosis and HIV sections, while the first part is more detailed, the second is written with broader strokes, referring more to the literature without going into details. In some cases, I felt the authors could have elaborated a little more instead of just referring to the literature. Two examples: rows 454-455, "an existing HIV infection can affect neutrophil function in different ways [155]", or rows 475-479, where it would be interesting to read more about what the strain-dependent beneficial and detrimental effects of type I interferons are exactly, and how do each of the various factors (such as stage of infection, host immune status and granuloma cytokine microenvironment) influence these effects.
2) Row 59-60, "infection within the macrophages and reinfection of newly arrived phagocytic cells...". The phrase reinfection is a little misleading, maybe could be just changed to "infection".
3) row 68, "a straightforwardly progress to TB may occurs."
4) row 93, "recombining activating gene (RAG)" correctly: recombination activating gene (RAG).
5) Figure 1 seems to be adapted from Figure 1 of reference [41], but the labeling on the resulting figure is less clear than the parent figure was. The disambiguation is missing for the 1R, 2R stars. It could be clarified by a line or an arrow, exactly which branch-segment of the phylogenetic tree should the boxes "750 million years ago" and "450 million years ago" point to. Images are included for some branches but not all, e.g. missing image for bony fish and amphibians, the reptile image is in between the amphibian and reptile branches. On the top right, the grey triangles that contain "innate immunity..." and "adaptive immunity..." usually signify a decrease/decline, so the reason for using that triangle shape there is misleading. What is the reason for showing the 2 branches for Cyclostomes and 2 branches for bony fishes if the figure doesn't discuss a difference between these branches? Since zebrafish is more recently evolved compared to the sturgeon, shouldn't the 2 bony fish branches be switched?
6) Row 117, The use of "therefore" seems unwarranted.
7) Row 166, 322, 427, 494, “PDL1” and “PD1” are correctly PD-L1 and PD-1.
8) Row 221, “Mtb infection evolved” either Mtb, or the host (or both co)evolved, but not the infection.
9) Row 234, “secretion” is doubled
10) Row 233-238 discusses the “another” effects of IL-1beta, but seems a little out of place. This section might better belong in row 217 after reference [22], which introduced IL-1beta and its protective effect.
11) Rows 243-245, duplicated sentence.
12) Figure 2, some text are too small to read, e.g. inside the mitochondrion and in the metabolic pathways.
13) Row 307, “10^5 fg/ml”, if this information is drawn from reference [117], figure 5J, then shouldn’t it be 10^4 fg/ml? The median and interquartile range of IFN-alpha2a concentration there seems to be around 10 pg/ml?
14) Rows 307-308, “UI/ml”, does this refer to international unit/ml? if so, shouldn’t it be IU/ml?
15) Row 310, “a signature of IFN-I consequences becomes evident”, please clarify.
16) Rows 339 and 340, instead of “impact” and impacts, please clarify impact in what direction, is it a positive or negative impact, e.g. induces / leads to / results in / contributes to, etc.
17) Figure 3, what is the blue cell in the right arm of the scale, a macrophage?
18) Row 379, “seems to occurs”
19) disambiguation of the abbreviations is missing from the legend of Figure 4.
20) Row 444, please add references to the statement.
21) Rows 446-447, might need rephrasing. “evolution” -> development? “straightforward” -> directly? “surpassing” -> bypassing?
22) Rows 465-466, “…the depletion of T-CD4+ lymphocytes induced by HIV [164], including TH1 and CTLs…” - CTLs are not CD4+ T lymphocytes.
23) Row 496, “clear and constant” -> Evident and sustained?
Comments on the Quality of English LanguageOnly minor editing of English language required.
Author Response
Reviewer1
The manuscript biomolecules-3088345 titled “Role of type I interferons during Mycobacterium tuberculosis and HIV infections” by Elsa Anes et al., is a review focused on summarizing the distinct roles of type I interferons in the acute and chronic stages of both tuberculosis and HIV infection, and importantly, the mechanisms by which type I interferons can contribute to Mtb infection in HIV-infected individuals and to disease progression during Mtb-HIV coinfection. The review draws insight from 176 references, including the latest works on the topic, it is clearly written, easy to read and to understand, and it represents a valuable contribution to the field. The previous reviews I have found, discussing the role of type I interferons in Mtb-HIV coinfection (du Bruyn et al., Current Opinion in HIV and AIDS 13(6):p 455-461, November 2018., DOI: 10.1097/COH.0000000000000501 and Ahmed et al., Oral Dis. 2016 Apr;22 Suppl 1:53-60. doi: 10.1111/odi.12390. PMID: 27109273.) highlight several mechanisms discussed in the current review but neither discusses the topic in such detail or contain works as recent as the manuscript of Anes et al. In conclusion, I recommend the acceptance of the current manuscript after minor revision.
1) I feel a slight break in the flow of the paper between the tuberculosis and HIV sections, while the first part is more detailed, the second is written with broader strokes, referring more to the literature without going into details. In some cases, I felt the authors could have elaborated a little more instead of just referring to the literature. Two examples: rows 454-455, "an existing HIV infection can affect neutrophil function in different ways [155]", or rows 475-479, where it would be interesting to read more about what the strain-dependent beneficial and detrimental effects of type I interferons are exactly, and how do each of the various factors (such as stage of infection, host immune status and granuloma cytokine microenvironment) influence these effects.
Re: Thank you for your suggestion. Further details have been provided on this topic
2) Row 59-60, "infection within the macrophages and reinfection of newly arrived phagocytic cells...". The phrase reinfection is a little misleading, maybe could be just changed to "infection".
Re: We corrected to “infection” as peer suggested
3) row 68, "a straightforwardly progress to TB may occurs."
Re: we corrected to: “In immunocompromised patients, following the primo infection a direct progress to TB may occurs, which can disable the latent phase of infection”.
4) row 93, "recombining activating gene (RAG)" correctly: recombination activating gene (RAG).
Re: thank you for the correction, that we introduced now.
5) Figure 1 seems to be adapted from Figure 1 of reference [41], but the labeling on the resulting figure is less clear than the parent figure was. The disambiguation is missing for the 1R, 2R stars. It could be clarified by a line or an arrow, exactly which branch-segment of the phylogenetic tree should the boxes "750 million years ago" and "450 million years ago" point to. Images are included for some branches but not all, e.g. missing image for bony fish and amphibians, the reptile image is in between the amphibian and reptile branches. On the top right, the grey triangles that contain "innate immunity..." and "adaptive immunity..." usually signify a decrease/decline, so the reason for using that triangle shape there is misleading. What is the reason for showing the 2 branches for Cyclostomes and 2 branches for bony fishes if the figure doesn't discuss a difference between these branches? Since zebrafish is more recently evolved compared to the sturgeon, shouldn't the 2 bony fish branches be switched?
Re: we address all these points in the present corrected figure 1
6) Row 117, The use of "therefore" seems unwarranted.
Re: we removed the word therefore from the statement.
7) Row 166, 322, 427, 494, “PDL1” and “PD1” are correctly PD-L1 and PD-1.
Re: thank you for the correction that we introduced in the present version
8) Row 221, “Mtb infection evolved” either Mtb, or the host (or both co)evolved, but not the infection.
Re: we removed “infection” from the phrase
9) Row 234, “secretion” is doubled
Re: we corrected that
10) Row 233-238 discusses the “another” effects of IL-1beta, but seems a little out of place. This section might better belong in row 217 after reference [22], which introduced IL-1beta and its protective effect.
Re: We think that if moved it will cut the flow on information on PAMPS, ESAT 6, inflammasome and IL1 beta production. We prefer to present as it is.
11) Rows 243-245, duplicated sentence.
Re: we removed the duplicated sentence from the present version.
12) Figure 2, some text are too small to read, e.g. inside the mitochondrion and in the metabolic pathways.
Re: we increased the size of the relevant molecules for a better visualization
13) Row 307, “10^5 fg/ml”, if this information is drawn from reference [117], figure 5J, then shouldn’t it be 10^4 fg/ml? The median and interquartile range of IFN-alpha2a concentration there seems to be around 10 pg/ml?
Re: 10^4 fg/ml is for healthy controls.10^5 fg/ml is the median for severity patients. This information comes from ref 117 in the previous version. The median concentration comes from reference 118 when comparing TB, LTB and influenza in this figure: https://www.frontiersin.org/files/Articles/471810/fcimb-09-00296-HTML/image_m/fcimb-09-00296-g001.jpg from the article https://doi.org/10.3389/fcimb.2019.00296 (ref118 now ref 206). We introduced that reference at the end of the statement.
14) Rows 307-308, “UI/ml”, does this refer to international unit/ml? if so, shouldn’t it be IU/ml?
Re: we corrected accordingly
15) Row 310, “a signature of IFN-I consequences becomes evident”, please clarify.
RE: We substituted for “However, an examination of the transcriptomic profile of ISGs reveals a signature of IFN-I in peripheral blood leukocytes, consequences becomes evident, highly correlated with the extend of lung pathology [204,207]. This allows the distinction between active disease and LTB [206]. These include the overexpression of IFITs, MX1, OAS1, IRF1, and ISG15 [204,207,208]. The downstream immune cellular responses to IFN-I signalling, including the overexpression of IL 10 and PD-L1, are at least in part mediated by type I IFN-activated STAT3 that leads to the repression of signals for proinflammatory cytokines [66] (Figure 2). The overexpression of ISG15 is observed in human monocytes during tuberculosis, and plays a role as cytokine that induces IL-10 secretion [208].”
16) Rows 339 and 340, instead of “impact” and impacts, please clarify impact in what direction, is it a positive or negative impact, e.g. induces / leads to / results in / contributes to, etc.
Re: we substituted “impact” by an “increase”
17) Figure 3, what is the blue cell in the right arm of the scale, a macrophage?
Re: In the left-hand square, near the weight balance, we illustrate a form of neutrophil NETosis. In the left-hand square all cells are neutrophils. In the right-hand square all cells are macrophages
18) Row 379, “seems to occurs”
Re: correction introduced
19) disambiguation of the abbreviations is missing from the legend of Figure 4.
Re: we introduced the list of abbreviations
20) Row 444, please add references to the statement.
Re: reference introduced
21) Rows 446-447, might need rephrasing. “evolution” -> development? “straightforward” -> directly? “surpassing” -> bypassing?
Re: we rephrase for” Furthermore, the effects of a high IFN-I long-lasting signalling induced by an established HIV infection may contribute to the rapid development of an Mtb primo infection directly to active TB, overcoming LTB.”
22) Rows 465-466, “…the depletion of T-CD4+ lymphocytes induced by HIV [164], including TH1 and CTLs…” - CTLs are not CD4+ T lymphocytes.
Re: we meant to state: including TH1 and consequently a depletion of CTLs that are dependent on the cytokines of the last for become effector cytotoxic cells. Both are crucial for controlling Mtb infection.
23) Row 496, “clear and constant” -> Evident and sustained?
Re: corrections introduced
Thanks for the accurate review of the manuscript
Reviewer 2 Report
Comments and Suggestions for Authors
The review by Anes E and colleagues presents a complete and precise update on the distinct role of type I IFN during these different stages of Mycobacterium tuberculosis (Mtb) infection, HIV infection and the co-infection. This review is well-written. This manuscript is based on an important topic and the authors have done well.
However, addressing the following issues can further improve the manuscript
Introduction: Introduction part needs to improve as the current format doesn’t blend all three together HIV, TB and interferon gamma. Please consider improving the introduction section. Please consider starting with the HIV infection first and then come to point how HIV induces interferon and then how HIV evolves to counter the effect of these immune responses. Further add how co-infection with tuberculosis can add complexity of HIV pathogenesis.
Figure fonts sizes are an issue here. Font’s size needs to increase.
Comments on the Quality of English LanguageCheck for few typo errors. Otherwise english are fine.
Author Response
Reviewer 2
Introduction: Introduction part needs to improve as the current format doesn’t blend all three together HIV, TB and interferon gamma. Please consider improving the introduction section. Please consider starting with the HIV infection first and then come to point how HIV induces interferon and then how HIV evolves to counter the effect of these immune responses. Further add how co-infection with tuberculosis can add complexity of HIV pathogenesis.
RE. The introduction merely provides a brief overview and general introduction; it does not delve into the relationship between the interferons. We have expanded the text to include additional information on how co-infection with TB can also contribute to the complexity of HIV pathogenesis. We agree with this suggestion and have rearranged the order of presentation, beginning with HIV infection. We included more details about HIV infection and have introduced the new figure 2 in response to the other reviewer’s suggestion of initially presenting IFN signaling for ISGs as an antiviral mechanism. It is therefore logical to alter the sequence and present first HIV infection, thus facilitating a more coherent flow of information.
Figure fonts sizes are an issue here. Font’s size needs to increase.
Re: we have increased the font sizes in all figures in the present version as peer suggested.
Reviewer 3 Report
Comments and Suggestions for Authors
In this review, the authors analyzed the function of IFN according to Mtb and HIV infection. The function and reversal of IFN and Mtb infection are appropriately described. In addition, we attempted to analyze the role of IFN in more detail by covering the action of IFN in HIV infection and Mtb and HIV co-infection. I think most of the content will be of interest to readers, but some minor revisions are needed.
1. Because the content is already so well known, the drawback is that there is a lack of new recognition.
2. It would be better to exclude the HIV infection part from the title as it does not cover much of the topic. It would be better to title the title itself Mtb and the function and role of IFN by co-infection.
3. It is believed that the quality of the review will improve if other co-infections are added such as HCV and HBC.
4. There is no explanation of the functions of proteins expressed by ISG stimulation. Since the role of IFN itself is to induce the expression of ISG, descriptions of at least some of these proteins should be added.
5. A description of the Mtb infection route and the immune mechanisms expressed during infection should also be provided.
6. It would be good to add a discussion on the use of IFN in relation to the treatment of Mtb.
Comments on the Quality of English LanguageMinor errors
Author Response
Reviewer 3
In this review, the authors analyzed the function of IFN according to Mtb and HIV infection. The function and reversal of IFN and Mtb infection are appropriately described. In addition, we attempted to analyze the role of IFN in more detail by covering the action of IFN in HIV infection and Mtb and HIV co-infection. I think most of the content will be of interest to readers, but some minor revisions are needed.
1. Because the content is already so well known, the drawback is that there is a lack of new recognition.
Re: A considerable amount of information is available on interferons during the active disease (tuberculosis) and during HIV infection. However, we felt that there was a lack of systematic and organized information considering acute infection, or chronic infection, particularly during latent infection with Mtb and during co-infection with HIV. This is the subject of the present manuscript. Furthermore, the existing literature lacked a clear schematic representation of the role and interception of all key players.
2. It would be better to exclude the HIV infection part from the title as it does not cover much of the topic. It would be better to title the title itself Mtb and the function and role of IFN by co-infection.
Re: We provide a more detailed information on HIV as pointed by the other reviewer providing more emphasis on this particular infection. Since the other two reviewers are not in line with the proposed title alteration, we maintain the title as it is. HIV as a recent pandemic and their relevance as pathogen and interferons interaction we decided to give him the right emphasis on the title and provide more detailed information on the present version of the manuscript.
Following the suggestion of the other reviewer, this version of the manuscript begins with the HIV infection after the analysis of the origin of interferons. We provide a new figure that presents some classes of ISGs responding to your suggestion. It is our hope that the modifications made will address your concerns.
3. It is believed that the quality of the review will improve if other co-infections are added such as HCV and HBC.
Re: The objective was to concentrate on two infections identified by the WHO as being particularly prevalent in syndemic conditions when they occur in co-infection.
4. There is no explanation of the functions of proteins expressed by ISG stimulation. Since the role of IFN itself is to induce the expression of ISG, descriptions of at least some of these proteins should be added.
Re. We totally agree with the reviewer, and we included that information in the part of interferons origin, in the present figure 2. We also included a more detailed explanation on HIV part when referring ISGs as antivirals.
5. A description of the Mtb infection route and the immune mechanisms expressed during infection should also be provided.
Re: we have already included the explanation of the immune responses to Mtb infection along the several stages of the infection from primo-infection, latent infection, and active disease. The innate immune response mediated by cytokines that are protective on the axis IL-12/TNF/IL1, and the evolution to the uncontrolled necrotic phase of the disease. All of these processes are connected to type I and II interferons. In response to the reviewer’s concerns, we have included a few sentences highlighted in the manuscript, highlighted in yellow, to emphasize the phases and immune mechanisms during Mtb infection. Indeed, we have highlighted in grey those immune mechanisms that were already included in the previous version. Furthermore, we cited previous reviews on the subject in which we provided a comprehensive analysis of those immune responses (i.e. Azevedo-Pereira et al 2023)
6. It would be good to add a discussion on the use of IFN in relation to the treatment of Mtb.
Re: A brief overview is provided on the potential applications of interferons for the treatment of TB and the contexts in which they can be used.